# Water Vapour Promotes CO$_2$ Transport in Poly(ionic liquid)/Ionic Liquid-Based Thin-Film Composite Membranes Containing Zinc Salt for Flue Gas Treatment

**Daria Nikolaeva** [1,2,*,†] **, Sandrine Loïs** [3] **, Paul Inge Dahl** [4] **, Marius Sandru** [4] **, Jolanta Jaschik** [5] **, Marek Tanczyk** [5] **, Alessio Fuoco** [6] **, Johannes Carolus Jansen** [6,*] **and Ivo F.J. Vankelecom** [1]

1  Membrane Technology Group (MTG), cMACS, Faculty Bio-science Engineering, Celestijnenlaan 200F, 3001 Leuven, Belgium; ivo.vankelecom@kuleuven.be
2  Materials & Process Engineering, UCLouvain, Place Sainte Barbe 2, 1348 Louvain-la-Neuve, Belgium
3  SOLVIONIC, Site Bioparc 195, route D'Espagne, BP1169, 31036 Toulouse CEDEX 1, France; sandrinelois@orange.fr
4  SINTEF Industry, Richard Birkelands vei 2B, 7034 Trondheim, Norway; PaulInge.Dahl@sintef.no (P.I.D.); marius.sandru@sintef.no (M.S.)
5  Institute of Chemical Engineering, Polish Academy of Sciences, ul. Bałtycka 5, 44-100 Gliwice, Poland; jjaschik@iich.gliwice.pl (J.J.); mtanczyk@iich.gliwice.pl (M.T.)
6  Institute on Membrane Technology, CNR-ITM, Via P. Bucci 17/C, 87036 Rende (CS), Italy; a.fuoco@itm.cnr.it
*  Correspondence: daria.nikolaeva@uclouvain.be (D.N.); johannescarolus.jansen@cnr.it (J.C.J.); Tel.: +32-10-472-473 (D.N.); +39-0984-492-031 (J.C.J.)
†  Current address: Materials & Process Engineering, UCLouvain, Place Sainte Barbe 2, 1348 Louvain-la-Neuve, Belgium.

**Abstract:** A poly(ionic-liquid) (PIL) matrix can be altered by incorporating additives that will disrupt the polymer chain packing, such as an ionic liquid (IL) and inorganic salts to boost their exploitation as materials for membrane production to be used in CO$_2$ capture. Herein, potential of PIL/IL/salt blends is investigated on the example of poly(diallyldimethyl ammonium) bis(trifluoromethylsulfonyl)imide (P[DADMA][Tf$_2$N]) with N-butyl-N-methyl pyrrolidinium bis(trifluoromethylsulfonyl)imide ([Pyrr$_{14}$][Tf$_2$N]) and zinc di-bis(trifluoromethylsulfonyl)imide (Zn[Tf$_2$N]$_2$). Composite material with IL and a higher amount of Zn$^{2+}$ showed an increase in the equilibrium CO$_2$ sorption capacity to 2.77 cm$^3$ (STP) cm$^{-3}$ bar$^{-1}$. Prepared blends were successfully processed into thick, dense membranes and thin-film composite membranes. Their CO$_2$ separation efficiency was determined using ideal and mixed-gas feed (vol% CO$_2$ = 50, dry and with 90% relative humidity). The dominant role of solubility in the transport mechanism is confirmed by combining direct gravimetric sorption measurements and indirect estimations from time-lag experiments. The maximum incorporated amount of Zn$^{2+}$ salts increased equilibrium solubility selectivity by at least 50 % in comparison to the parent PIL. All materials showed increased CO$_2$ permeance values by at least 30% in dry conditions, and 60 % in humidified conditions when compared to the parent PIL; the performance of pure PIL remained unchanged upon addition of water vapor to the feed stream. Mixed-gas selectivities for all materials rose by 10 % in humidified conditions when compared to dry feed experiments. Our results confirm that the addition of IL improves the performance of PIL-based composites due to lower stiffness of the membrane matrix. The addition of Zn$^{2+}$-based salt had a marginal effect on CO$_2$ separation efficiency, suggesting that the cation participates in the facilitated transport of CO$_2$.

**Keywords:** flue gas; poly(ionic liquid); $CO_2$ transport; thin-film composites; relative humidity; zinc; polymeric membranes

## 1. Introduction

Poly(ionic liquid)-based membranes for $CO_2$ capture overcome the possible problems of ionic liquid (IL) leaching from the supported IL membranes [1]. However, poly(ionic-liquids) (PILs) demonstrate a considerable reduction of $CO_2$ sorption capacity and diffusion rates due to higher stiffness of the polymer matrix, leading to lower $CO_2$ transport [2,3]. PIL matrix can be altered by incorporating additives that will disrupt the polymer chain packing, such as ILs and inorganic salts [4,5]. Poly(ionic liquid)s (PILs) have recently prompted extensive practical aspirations due to their potential to combine the large $CO_2$ sorption capacity of ionic liquids (IL) and good processability of conventional polymers [6–8]. PIL synthesis so-far focused mainly on the development of tailor-made materials from polymerisable IL-monomers, offering excellent gas separation properties [9–12]. The IL monomers should have a high degree of purity, and undergo elaborate synthetic procedures to produce large amounts of PILs with sufficient molecular chain length, which might present a critical factor in their further employment as selective membranes for $CO_2$ capture [13].

A more limited body of work explores the functionalization of conventional polymers used in membrane-based gas separation, such as cellulose acetate [14–16], poly(vinylbenzyl chloride) [17,18], polybenzimidazole [19–21], polyurethane [22] and poly(diallyldimethyl chloride)(P[DADMA] [Cl])[10,13,16,23–25], as alternative substrates for PIL-like structures. In contrast to IL-monomer polymerization, this route proposes the direct incorporation of IL-moieties in the polymeric backbone and/or further ion exchange. As P[DADMA][Cl] allows a simpler PIL or polyelectrolyte (PE) synthesis by only anion exchange, many existing studies focused on understanding the underlying phenomena connecting the physico-chemical-structural properties with performance evaluation, in which the group of Marrucho plays a pivotal role [10,13,24–29]. However, the P[DADMA][Cl] PIL derivatives often demonstrate low mechanical strength and $CO_2$ sorption capacity [13,28], requiring the conditioning of synthesised PILs with plasticizing agents or pure ILs that would act as a plasticiser for the rigid polymer chains. In this case, IL also acts as an additional sorption phase for permeating $CO_2$ molecules [11]. Additionally, the low viscosity of IL segments would allow faster gas transport due to enhanced diffusivity [30].

Several critical reviews report on the progress in PIL/IL composite material development [5,6,31]. Several PIL/IL composites excel and cross the Robeson plot for $CO_2/N_2$ separation [2,10,26,28,32]. However, the majority of them exhibit moderate $CO_2$ permeabilities (10–300 Barrer) and $CO_2/N_2$ selectivities (20–40) [28,29]. Those are still far below the industrially relevant $CO_2$ permeance of 1000 GPU and $CO_2$ selectivity of at least 50 [33]. Moreover, while recent scientific reports underscore the improved separation performance of IL-based gas separation membranes in dry feed, real flue gas always contains water vapor, making the investigations in humidified conditions obligatory [17,34,35].

The PIL/IL composite's performance was further altered by the addition of copper salt to facilitate the $CO_2$ transport by Zarca et al. [4]. In the present work, zinc ions are introduced to investigate their influence on the performances of PIL/IL composites. In nature, $Zn^{2+}$ participates in the chemical binding of $CO_2$ molecules, as a structural component of more than 300 enzymes, with carbonic anhydrase being the most widely known example [36–38]. $Zn^{2+}$ is at the center of a catalytically active site with enhanced Lewis acidity in tetrahedral configuration, coordinating three ligands and a water molecule [39]. The catalytic nature of $Zn^{2+}$ has been recently applied in the fields of $CO_2$ valorization and absorption [40,41], while the application in membrane-based gas separation remains limited [42–44]. However, the reversible nature of $Zn^{2+}$ catalyzed reactions may be further exploited in $CO_2$ selective membranes through the expected improvement of the affinity of $CO_2$ molecules to $Zn^{2+}$ binding sites. While high catalytic activity of $Zn^{2+}$ often requires the usage of halogen salts [45,46],

their strength might be excessive, resulting in saturation of the membrane with the reaction products or too strong a coordination of $CO_2$. Thus, anions with lower catalytic activity would be more suitable for membrane gas separation: $CO_3^{2-}$, $NO_3^-$, $OAc^-$, $OTf^-$, $[Tf_2N]^-$, etc. Drawing on the existing literature, $[Tf_2N]^-$ has already received wide approval for its high $CO_2$ affinity, arising from the presence of fluorine atoms [47–50]. Additionally, it has much higher chemical stability in comparison to other fluorine-containing anions, such as $[BF_4]^-$ and $[BF_6]^-$, which hydrolyze in ambient conditions forming hydrofluoric acid [51].

This work implements $Zn[Tf_2N]_2$ salt in PIL/IL composites with the expectation to intensify the $CO_2$ sorption selectivity in PIL-based materials (Figure 1). The sorption performance of developed materials is assessed both in direct and indirect measurements by gravimetric sorption and time-lag experiments, respectively. Furthermore, this paper highlights the possibility to deploy the PIL/IL/$Zn^{2+}$ composite materials as competitive TFC membranes for robust $CO_2$ capture from feeds similar to flue gas with substantial humidity content.

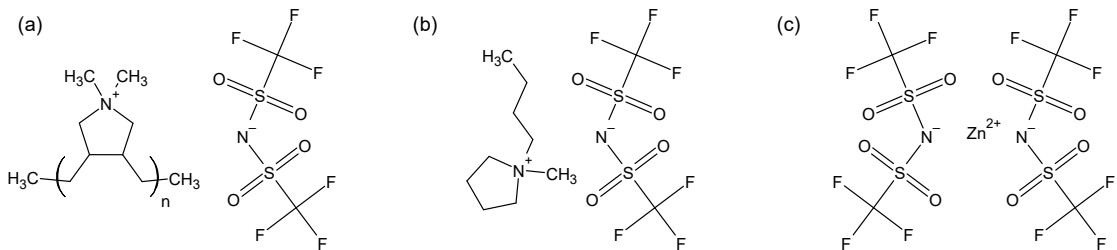

**Figure 1.** Chemical structures of PIL/IL/$Zn^{2+}$ composite constituents: (**a**) poly-(diallyldimethyl ammonium) bis(trifluoromethylsulfonyl)imide (P[DADMA][$Tf_2N$]); (**b**) N-butyl-N-methyl pyrrolidinium bis(trifluoromethylsulfonyl)imide ([$Pyrr_{14}$][$Tf_2N$]) and (**c**) zinc di-bis(trifluoromethylsulfonyl)imide ($Zn[Tf_2N]_2$).

## 2. Materials and Methods

### 2.1. Materials

N-butyl-N-methyl pyrrolidinium bis(trifluoromethylsulfonyl)imide ([$Pyrr_{14}$][$Tf_2N$], 99.9 %, $M_w$ = 422.41 $g\,mol^{-1}$), poly(diallyldimethyl ammonium) bis(trifluoromethylsulfonyl)imide (P[DADMA][$Tf_2N$], 99.9 %, $\overline{M}_n$ = 275000 $g\,mol^{-1}$) and zinc bis(trifluoromethylsulfonyl)imide ($Zn[Tf_2N]_2$, 99.5 %, $M_w$ = 625.68 $g\,mol^{-1}$) were provided by SOLVIONIC under the trade names Pyr0408a, Amsf5408a and M3008c, respectively. All chemicals were used as received, handled concerning the hazard and precautionary statements (warning): H315, H319, H335, P280, P305+ P351+ P338, P310.

Polypropylene/polyethylene (PP/PE) non-woven fabric Novatexx® 2483 was received from Freudenberg (Germany), and Matrimid® 9725 was kindly provided by Huntsman (Switzerland). P-xylylenediamine (XDA, > 98 %) cross-linker was acquired from Fluka. Polydimethylsiloxane (PDMS, two-component RTV615 silicone rubber compound kit) was purchased from Momentive Performance Materials (Leverkusen, Germany). Methanol (Acros, 99.8 %), N-methylpyrrolidinone (NMP, Acros, 99 %), tetrahydrofuran (THF, Acros, 99.5 %), acetone (Merck, 99.8 %), n-hexane (Merck, 99 %), ethanol (Fisher Scientific, 99.5 %) and isopropanol (VWR, 99.5 %) were all used without further purification.

Carbon dioxide ($CO_2$, Messer, 99.995 %), nitrogen ($N_2$, Air Products, 99.9999 %) and helium (He, Air Products, 99.9999 %) were used for gravimetric gas sorption measurements as received. The gases used in the time-lag tests (nitrogen, oxygen, methane, helium, hydrogen and carbon dioxide, all 99.99+%) were supplied by Sapio, Italy.

## 2.2. Composite PIL/IL/Zn$^{2+}$ Materials

Composite materials were prepared by solution blending of poly(diallyldimethyl ammonium) bis(trifluoromethylsulfonyl)imide (P[DADMA][Tf$_2$N]) or PIL, N-butyl-N-methyl pyrrolidinium bis(trifluoromethylsulfonyl)imide ([Pyrr$_{14}$][Tf$_2$N]) or IL, and salt zinc[bis(trifluoromethylsulfonyl)imide]$_2$ (Zn[Tf$_2$N]$_2$), with the components added in different proportions. Table 1 contains the labels assigned to the derived composite materials. The detailed procedure is reported in the supporting information.

**Table 1.** Composite PIL/IL/Zn$^{2+}$ materials.

| Sample | Composition | | | Ratios | | | Label |
|---|---|---|---|---|---|---|---|
| [Name] | PIL [wt %] | IL [wt %] | Zn$^{2+}$ Salt [wt %] | PIL [–] | IL [–] | Zn$^{2+}$ Salt [–] | [–] |
| PIL | 100 | – | – | 9 | 0 | 0 | P$_9$IL$_0$Zn$_0$ |
| PIL/IL | 60 | 40 | – | 9 | 6 | 0 | P$_9$IL$_6$Zn$_0$ |
| PIL/IL/Zn[Tf$_2$N]$_2$ ratio 9:1 | 56.3 | 37.5 | 6.2 | 9 | 6 | 1 | P$_9$IL$_6$Zn$_1$ |
| PIL/IL/Zn[Tf$_2$N]$_2$ ratio 1:1 | 37.5 | 25.2 | 37.3 | 9 | 6 | 9 | P$_9$IL$_6$Zn$_9$ |

## 2.3. Material Characterization

### 2.3.1. Thermo-Gravimetric Analysis (TGA)

TGA was performed using a STA 449 C Jupiter$^{®}$ TG-DTA analyzer (Netzsch-Gerätebau, Germany) between 25 °C to 700 °C under nitrogen atmosphere and a heating rate of 10 °C min$^{-1}$. Prior to the TGA measurements, the samples were dried in vacuum oven at 60 °C for 24 h to eliminate the absorbed moisture.

### 2.3.2. Differential Scanning Calorimetry (DSC)

The thermal properties of the membranes were determined by a Pyris Diamond Differential Scanning Calorimeter (Perkin Elmer, USA) equipped with an intra-cooler refrigeration system. Samples of 10–15 mg were wrapped in a small disk of aluminium foil ($< 5$ mg) and were subjected to a heating/cooling/heating cycle in the range from $-50$ to 250 °C at a rate of 15 °C min$^{-1}$. An empty sample holder was used as a reference. Baseline subtraction was used to reduce the curvature of the baseline. Temperature and heat flow were calibrated with Indium and Zinc standards. The melting points and the enthalpies for indium ($T_{mp} =$ 156.6 °C, $\delta_{Hm} =$ 28.5 J g$^{-1}$) were used for the calibration of temperature and heat capacity, respectively.

### 2.3.3. Sorption Analysis

CO$_2$ and N$_2$ sorption isotherms were obtained at 20 °C using a gravimetric analyzer (Hiden Isochema IGA-003, UK) with a resolution of 0.2 µg and the buoyancy force correction (Archimedes principle) [52–54]. The chosen experimental CO$_2$ pressure range was 0–5 bar for all samples, and the N$_2$ pressure range was 0–5 bar for **P$_9$IL$_0$Zn$_0$** and **P$_9$IL$_6$Zn$_9$**, and 0–18 bar for **P$_9$IL$_6$Zn$_0$** and **P$_9$IL$_6$Zn$_1$**. Samples were degassed under vacuum at 70 °C for 10–70 h before the measurements. The time required to obtain each experimental value equalled 360 min for all samples. A detailed explaination of the methodology is reported in the electronic supporting information.

*2.4. Membrane Preparation*

2.4.1. Self-Standing Dense Membranes

The neat PIL-based membranes were prepared using a solution casting method reported previously [55]. PIL or PIL-based composite material was dissolved as an 8 wt % solution in acetone. After homogenization and de-gassing, the polymer solution (2 mL) was cast onto a polyester film (76.2 µm packaging grade), firmly fixed to the stainless steel frame ($\oslash$ 50 mm), in a controlled environment at $25 \pm 1$ °C and $20 \pm 1\%$ relative humidity (RH). The polymeric film was left to dry for ca. 72 h. The dry membranes were removed from the frame and peeled from the support. Their smooth shiny surface and transparent bulk suggested good compatibility of the different components and the formation of a homogeneous mixture at the molecular level or a physical blend with a maximum domain size well below the wavelength of visible light. Thicknesses of the self-standing dense membranes were measured by electronic micrometre on ten positions for each membrane prepared and was equal to $48 \pm 2.82$ µm over all samples prepared. The membranes were additionally dried in the vacuum chamber for 3 h prior to time-lag measurements.

2.4.2. Thin Film Composite (TFC) Membranes

PIL-based coatings were physically deposited onto polyimide (PI) nano-filtration membranes playing the role of a mechanical support yielding thin-film composite (TFC) membranes. PI supports were prepared using a phase inversion method from 15 wt % solution of Matrimid® 9725 in a mixture of solvent containing NMP, THF and $H_2O$ in 62.25 wt %, 20.75 wt % and 2.00 wt %, respectively. The obtained homogeneous polymer solution was filtered, degassed and cast on the Novatexx® 2483 non-woven fabric with a doctor blade positioned at 200 µm. After 30 s of preliminary solvent evaporation, the non-woven with a PI coating was submerged into a bath filled with deionised water to precipitate the PI film [56]. The PI supports were further cross-linked in 0.63 wt % XDA solution in methanol for 3 d, rendering them insoluble in common solvents [57].

All PIL-based materials were dried under vacuum overnight at 60 °C to eliminate the absorbed moisture. Coating solutions were obtained by dissolution of the active polymer in acetone to acquire a final concentration of 4 wt%. The solutions were magnetically stirred to yield a homogeneous phase, filtered and allowed to degas overnight to avoid defect formation.

The composite material solutions were spray coated onto the PI supports using an Exactacoat ultrasonic coating system (Sono-Tek Corp) with an AccuMist ultrasonic nozzle [58]. To obtain smooth, thin, and defect free TFC membranes, the 4 wt % solution of PIL and PIL/IL blends in acetone was sprayed with 4.0 W power output on the ultrasonic nozzle and a spray flow rate of 1 mL min$^{-1}$ was applied, corresponding to a coverage of 20 µL cm$^{-2}$. For each membrane, 4 layers were coated, with a subsequent drying at room temperature for 10 min between each layer. The samples were finally dried/cured at room temperature. A thin layer of PDMS was applied over the PIL-based selective layer as a final stage of the membrane preparation, by coating with an 10 wt% solution in hexane, following the procedure suggested by the supplier and described elsewhere [16].

Thicknesses of TFC membranes were measured using the manufacturer's SEM software for each layer separately. To obtain average thickness of a layer, the measurement was conducted on at least three identical membranes on five spots for each membrane observed. The selective layer thickness ranged between $2.9 \pm 0.1$ to $9.8 \pm 0.1$ µm over all samples prepared [17].

*2.5. Membrane Performance Evaluation*

2.5.1. Time-Lag Measurements

Single gas permeability and time-lag experiments were performed on a fixed volume/pressure increase instrument constructed by Elektro and Elektronik Service Reuter (Geesthacht, Germany)

on circular samples with an effective area of $11.3 \, cm^2$ or $2.14 \, cm^2$. The feed gas was set at 1 bar for all the gases, and measurements at lower pressures (i.e. 0.8 bar, 0.6 bar, 0.4 bar, 0.2 bar and 0.1 bar) were performed only for $CO_2$ in order to analyze the pressure dependence. The permeate pressure was measured up to 13.3 mbar with a resolution of 0.0001 mbar. The gases were always tested in the following order: $H_2$, He, $O_2$, $N_2$, $CH_4$ and $CO_2$, and the effective degassing was guaranteed by a turbo molecular pump (Pfeiffer Vacuum). Permeabilities ($P_i$) are reported in Barrer (1 Barrer $= 10^{-10} \, cm^3 \, (STP) \, cm \, cm^{-2} \, s^{-1} \, cmHg^{-1}$), and the diffusion coefficient was calculated from the so-called permeation time lag, $\Theta$ (s). The ratio of the permeability over the diffusion coefficient gives the gas solubility coefficient in its approximate form. A more detailed description of the method can be found elsewhere [59,60].

### 2.5.2. Scanning Electron Microscopy (SEM)

The TFC morphology of the membranes was investigated using a Hitachi N-3400 SEM, Japan with a set acceleration voltage of 15 kV. Samples for SEM analysis were obtained by fracturing the dry quick-frozen in liquid nitrogen membrane segments and sputtering them with gold.

### 2.5.3. Mixed-Gas Separation

Mixed gas permeation tests were performed on the in-house built separation unit at SINTEF (Norway) to investigate the influences of the process parameters (feed pressure, temperature and relative humidity) [61]. This setup provides a high degree of automation, enabling continuous data collection (temperature, pressure, gas flow rate, composition and humidity) during operation under process conditions close to real flue gas treatment. The membranes with an active area of $19.63 \, cm^2$, were tested using synthetic flue gas: mixture of 15 mol % $CO_2$ and 85 mol % $N_2$ with and without 90% of relative humidity (RH). To verify the reproducibility of the results obtained, at least three replicas of TFC membranes were tested under the same conditions. The results are presented as average values with standard deviations indicated by error bars. The permeate gas was continuously withdrawn with He as sweep gas, and sent to a gas chromatograph and a flow meter for the permeate composition and flow rate measurements, respectively. The sweep gas stream was additionally humidified in the experiments with wet feed gas. The online composition analysis of the permeate and feed streams was performed by a micro-gas chromatograph (GC, Agilent, USA). A detailed explanation of the methodology is reported in the electronic supporting information.

## 3. Results and Discussion

### 3.1. Thermoanalysis of Composite PIL/IL/Zn$^{2+}$ Materials

Figure 2 depicts the decomposition curves of the composite PIL-based materials in comparison to the reference material P[DADMA][Tf$_2$N], referred to as **P₉IL₀Zn₀** [16]. PIL/IL/Zn$^{2+}$ composite materials demonstrate worsening thermal stability with increasing addition of Zn[Tf$_2$N] salt. The initial mass loss fluctuates in the range between $300 \, °C$ for **P₉IL₆Zn₉** and $350 \, °C$ for **P₉IL₀Zn₀, P₉IL₆Zn₀** and **P₉IL₆Zn₁**.

For the pure PIL and PIL/IL composite, the major loss occurs at $430 \, °C$ with a rate of 2.5 % min$^{-1}$, corresponding to de-polymerization and consequent decomposition of separate IL moieties, starting from approximately $405 \, °C$, as reported elsewhere for [Pyrr$_{14}$][Tf$_2$N] [62]. The PIL/IL/Zn$^{2+}$ composite's decomposition curve decreases by $20 \, °C$ to $410 \, °C$ with **P₉IL₆Zn₁**, showing more thermal stability than **P₉IL₆Zn₉**. These results align well with the composite materials consisting of similar building blocks; namely, ILs and Zn[Tf$_2$N]$_2$ [63].

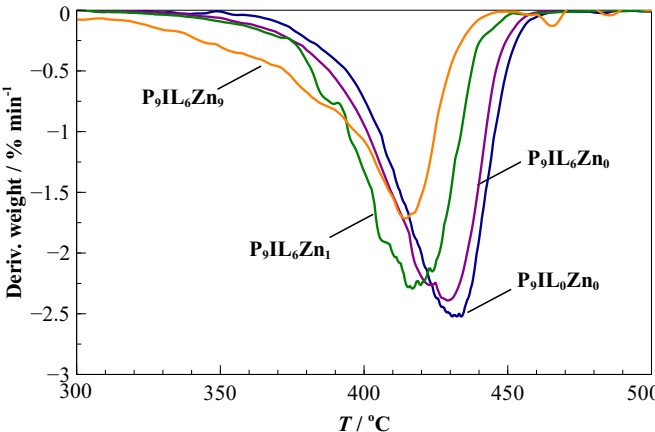

**Figure 2.** TGA spectra of $P_9IL_0Zn_0$, $P_9IL_6Zn_0$, $P_9IL_6Zn_1$ and $P_9IL_6Zn_9$ as functions of temperature.

DSC analysis shows extremely weak thermal effects in all samples. None of the samples have any evident primary transitions (melting point) in the temperature range from $-40$ to $180\,°C$ (ESI Figure S1). The extremely broad apparent peak in the first heating run (ESI Figure S1a) is probably associated with different responses of the membranes related to their manufacturing, i.e., enthalpy relaxation, and to evaporation of traces of humidity or residual solvent [64,65]. The neat $P_9IL_0Zn_0$ and $P_9IL_6Zn_0$ show a faintly visible glass transition temperature near $80\,°C$ and $20\,°C$, in close agreement with the literature data for the same polymer [13]. This further descends to ca. $5\,°C$ for $P_9IL_6Zn_9$. Although the DSC curve of pure $[Pyrr_{14}][Tf_2N]$ was not measured experimentally, the available literature reports its $T_g$ at $-81\,°C$ [66–68]. The $T_g$ of the PIL/IL blends, $T_{g,blend}$, which can be estimated for example by the simple Fox-Equation (Equation (1)), is expected to decrease rapidly with increasing IL content.

$$\frac{1}{T_{g,blend}} = \frac{w_{PIL}}{T_{g,PIL}} + \frac{w_{IL}}{T_{g,IL}} \tag{1}$$

where $w_{PIL}$ and $w_{IL}$ are the weight fractions of the PIL and the IL, respectively, and $T_{g,PIL}$ and $T_{g,IL}$ are the glass transition temperatures of the pure components. The latter supports well the qualitatively more flexible behavior of the sample, confirming the plasticizing effect of the low molar mass IL.

*3.2. Sorption Behaviour*

Figure 3 depicts the $CO_2$ (a) and $N_2$ (b) sorption isotherms of PIL and PIL-based composite materials. The $CO_2$ sorption capacities lie within a relatively narrow range for all samples. This behavior is related to the chemical composition of the constituents: identical cationic moieties in $P[DADMA][Tf_2N]$ and $[Pyrr_{14}][Tf_2N]$, and $[Tf_2N]^-$ anion in PIL, IL and salt.

The $CO_2$ sorption isotherms are slightly non-linear in the low pressure region (0–2 bar) and become roughly linear for higher pressures, suggesting a weak dual-mode sorption mechanism (Figure 3a). To describe the sorption behavior of single gases ($CO_2$, $N_2$), experimental sorption isotherms were analyzed and fitted to the dual-mode sorption model (DMM) using non-linear regression (Table 2). However, the estimated values ought to be treated with care, as in the present case, only few points are available in the curved sections of the isotherms.

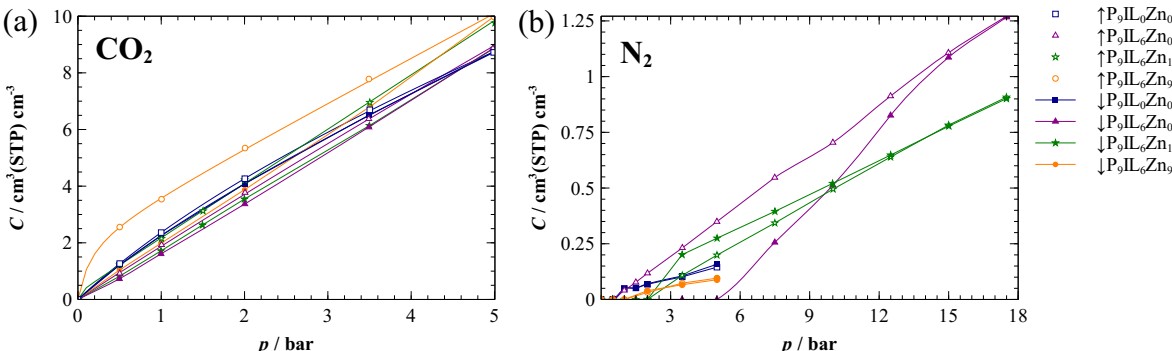

**Figure 3.** $CO_2$ (**a**) and $N_2$ (**b**) sorption isotherms in composites at 20 °C measured directly by gravimetric apparatus. Filled and open symbols denote absorption and desorption runs, respectively. The corresponding coloured solid curves provide guides for the eyes. In the case of (**a**), the curves are fitted with DMM.

**Table 2.** Dual-mode model parameters and separation properties obtained at 20 °C

| Sample Form | P9IL0Zn0 Powder | P9IL6Zn0 Solid Gel Cube | P9IL6Zn1 Solid Gel Cube | P9IL6Zn9 Solid Gel Cube |
|---|---|---|---|---|
| Particle radius ($r_{v0.5}$) (eq.)[a], μm | 25.4 | 3310 | 3090 | 2820 |
| Density ($\rho$), $g\,cm^{-3}$ | 1.542 | 1.448 | 1.494 | 1.619 |
| $CO_2$[b] 20 °C   $k_D$[c], $cm^3$ (STP) $cm^{-3}\,bar^{-1}$ | 1.12 | 1.78 | 1.84 | 1.79 |
| $C'_H$, $cm^3$ (STP) $cm^{-3}$ | 5.31 | 0 | 0 | 1.08 |
| $b$, $bar^{-1}$ | 0.29 | 0 | 0 | 9.8 |
| $S$[c], $cm^3$ (STP) $cm^{-3}\,bar^{-1}$ | 2.31 | 1.78 | 1.84 | 2.77 |
| $D$, $cm^2\,s^{-1}$ | $7.60 \cdot 10^{-9}$ | $1.03 \cdot 10^{-6}$ | $9.08 \cdot 10^{-7}$ | $5.19 \cdot 10^{-7}$ |
| $P$[d], Barrer | 2.32 | 245.5 | 222.7 | 191.6 |
| $N_2$[a] 20 °C   $k_D$[c], $cm^3$ (STP) $cm^{-3}\,bar^{-1}$ | 0.031 | 0.067 | 0.052 | 0.019 |
| $C'_H$, $cm^3$ (STP) $cm^{-3}$ | 0 | 0 | 0 | 0 |
| $b$, $bar^{-1}$ | 0 | 0 | 0 | 0 |
| $S$[d], $cm^3$ (STP) $cm^{-3}\,bar^{-1}$ | 0.031 | 0.067 | 0.052 | 0.019 |
| $D$, $cm^2\,s^{-1}$ | $2.30 \cdot 10^{-8}$ | $1.21 \cdot 10^{-6}$ | $8.90 \cdot 10^{-7}$ | $1.40 \cdot 10^{-6}$ |
| $P$[e], Barrer | 0.095 | 10.81 | 6.17 | 3.56 |
| $P_{CO_2} / P_{N_2}$[d] | 24.4 | 22.7 | 36.1 | 53.8 |

[a] The composite materials were thick solid gel discs that were cut into squares varying in size from 4.43 mm to 5.39 mm with equivalent radii given in the table. Therefore, $r_{v0.5}$ refers only to the pure poly(ionic-liquid) (PIL) sample. [b] Average relative error of 4.7 %. [c] As all experimental data points lay in the non-linear sorption region the accuracy in the determination of $k_D$ may be compromised. [d] At 1 bar. [e] Permeability was calculated indirectly from solubility $S_i$ and diffusion $D_i$ as product $P_i = S_i \cdot D_i$. Barrer = $10^{-10}\,cm^{-3}$ STP cm $cm^{-2}\,s^{-1}\,cmHg^{-1}$.

While the $CO_2$ sorption isotherm for pure **P9IL0Zn0** correlates well with the values calculated according to the traditional DMM sorption [16], the composite materials exhibit hysteresis between sorption and desorption runs with the degree of hysteresis increasing in the following order: **P9IL6Zn0** < **P9IL6Zn1** < **P9IL6Zn9**. The presence of hysteresis relates to the micropores that are not connected and are inaccessible to the gas at the low relative pressures [69]. As the $CO_2$ sorption in **P9IL6Zn0** approximates the linear Henry's sorption model, a significantly lower fractional free volume (FFV) is suggested, probably occupied by IL [70]. The absence of glass transition temperature also supports this hypothesis (ESI Figure S1c). Moreover, the stronger hysteresis revealed by $Zn^{2+}$-containing samples suggests the reversible chemisorption between these composites and $CO_2$ (Table 2).

Although **P9IL6Zn9** showed the broadest hysteresis for $CO_2$ sorption, we used the average $CO_2$ sorption values to estimate the DMM parameters based on the fact that hysteresis was closed at higher pressures. Insufficient equilibration time could affect the $CO_2$ sorption data for the sample with high affinity towards the $CO_2$ molecules. Our hypothesis rests upon the fact that the large concentrations of $[Tf_2N]^-$, known for its strong affinity for interactions with $CO_2$ [10,71], and $Zn^{2+}$, known for its

complexing nature and ability to coordinate $CO_2$ molecules [44], might lead to longer desorption times due to $CO_2$ chemisorption in/from the material.

The total $N_2$ sorption capacitywas very low (less than 0.1 wt % at 10 bar) in all samples investigated and decreased in the order $P_9IL_6Zn_0 > P_9IL_6Zn_1 > P_9IL_0Zn_0 > P_9IL_6Zn_9$. This difference in comparison to $CO_2$ is due to the far worse condensability of $N_2$ compared to $CO_2$ [72,73]. Therefore, one would expect all $N_2$ isotherms to follow Henry's sorption model. However, this trend deviated for $P_9IL_6Zn_0$ and $P_9IL_6Zn_1$ $N_2$ sorption isotherms, as strong hysteresis was observed in the lower pressure region ($< 9$ bar). Though the results for the desorption run in $P_9IL_6Zn_0$ and the adsorption run in $P_9IL_6Zn_1$ align well with the calculated values of DMM. These anomalies give an impression that some measurement points at lower pressure range might have had insufficient time to equilibrate, affecting their accuracy. However, the reason for this unusual curve shape is not completely clear, and is most likely related to a low measurement sensitivity or a low signal-to-noise ratio of the sorption instrument at low pressures and at very low amounts of sorption. There is no likely physical mechanism to explain such a curve shape, and calculations of the characteristic times based on the particle size and diffusion coefficient indicate that the total measurement time is sufficient to reach equilibrium. Nevertheless, the desorption curves should give a reliable range of the sorption behavior.

Overall, the solubility selectivity decreases with increasing gas pressure and reaches steady state at the maximum sorption capacity of the material (Figure 4). The $CO_2$ and $N_2$ adsorption in the FFV positively affect the $CO_2/N_2$ solubility selectivity of $P_9IL_6Zn_9$ in the low pressure region ($< 2$ bar). Additionally, Langmuir's sorption contribution enhances the solubility selectivity for $P_9IL_6Zn_9$ by at least 40 % at a higher pressure range ($>2$ bar) in the region controlled by Henry sorption model (Table 2). Similar behavior is observed for the pure PIL $P_9IL_0Zn_0$ which reveals a smaller contribution of Langmuir's sorption. The likely diminishing of the FFV content in the $P_9IL_6Zn_0$ and $P_9IL_6Zn_1$ leads to approximately 30 % and 50 % lower $S_{CO_2}/S_{N_2}$ for $P_9IL_6Zn_0$ and $P_9IL_6Zn_1$, respectively. Additionally, their performances in terms of $\alpha_S$, $\alpha_D$ and $\alpha_P$ remained nearly constant over the whole range of pressures investigated.

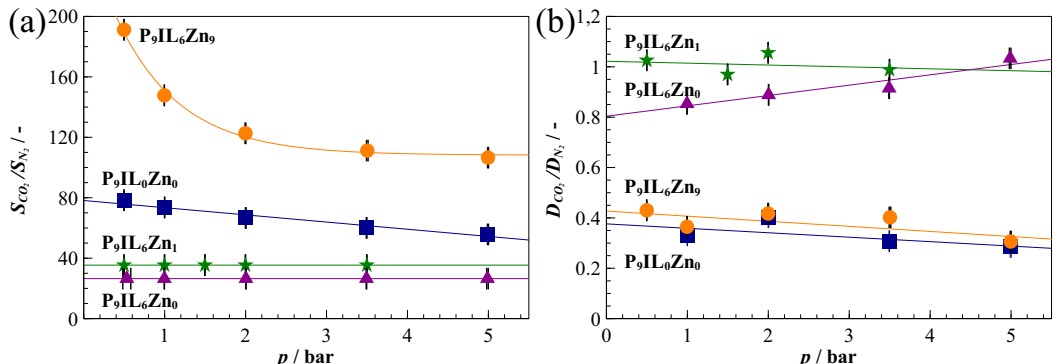

**Figure 4.** $CO_2/N_2$ ideal solubility (**a**) and diffusivity (**b**) selectivities derived from single gas sorption experiments at 20 °C. The solid curves provide guides for eyes.

The diffusivity selectivity remains nearly constant with increasing gas pressure for all membranes. $P_9IL_6Zn_0$ and $P_9IL_6Zn_1$ show almost no $D_{CO_2}/D_{N_2}$, as the composite material contains more than 37.5 wt % of IL liquid (Table 1). The plasticization of the polymer by the IL favors the diffusivity of both gases through the membrane. Although the $P_9IL_6Zn_9$ contains about 25 wt % of IL (Table 1), it has similar diffusion selectivity to the pure PIL ($P_9IL_0Zn_0$). This phenomenon indicates the excess amount of $Zn[Tf_2N]_2$ within the composite disrupts the IL domains and hinders the diffusion. Normally, $N_2$ diffusion is somewhat slower than $CO_2$ diffusion, as a result of its slightly larger effective diameter. Only in rubbers or soft materials is $CO_2$ diffusion much closer to that of $N_2$ or even up to ca. 30 % slower than $N_2$. Based on the DSC results and the glassy state of $P_9IL_0Zn_0$, the low $CO_2/N_2$ diffusion

selectivity is opposed to what one would expect [74] and further strengthens the hypothesis of $CO_2$ chemisorption in the PIL, or otherwise shows interaction, which may restrict diffusion of $CO_2$ [75].

Figure 5 depicts permeability selectivity ($P_{CO_2}/P_{N_2}$) as the product of ideal solubility and diffusivity selectivities (ESI Equation (8)). The permeability selectivity predicts considerably higher values for the composites containing $Zn^{2+}$ salt in comparison to PIL and PIL/IL materials. Nevertheless, the predicted separation performance also suggests that changes in the feed gas composition could have stronger effect on the composites with high Langmuir sorption component (**P9IL6Zn9** > **P9IL0Zn0**). The estimated indirect permeability selectivity and $CO_2$ permeability are consistent with the theoretical framework of the composite membranes containing ILs and inorganic additives [44,49,76], allowing the comparison of separation properties with other materials (ESI Figure S2). As expected, the PIL/IL/$Zn^{2+}$ composite materials exhibit higher $CO_2$ separation performance than the pure PIL, providing a strong incentive for their further direct assessment in the form of membranes for $CO_2$ capture.

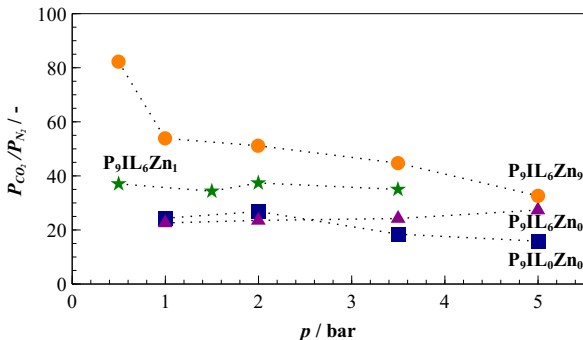

**Figure 5.** $CO_2/N_2$ permeability selectivity calculated from single gas sorption experiments at 20 °C. The black dotted curves provide guides for eyes.

### 3.3. Ideal CO₂ Separation Performance

The gravimetric sorption results demonstrate the materials with a strong affinity for $CO_2$ and increased $CO_2$ permeabilities. However, the accuracy of the gravimetric sorption data decays at pressures below 1 bar, resulting in lower reliability of the performance parameters that are calculated indirectly: $D_i$ and $P_i$.

ESI Figure S3 demonstrates a similar trend in $CO_2$ solubility between the investigated materials if we compare it to the adsorption runs of the gravimetric sorption measurements. The $CO_2$ sorption capacity decreases in the following order: **P9IL0Zn0** > **P9IL6Zn1** > **P9IL6Zn9** > **P9IL6Zn0**. This trend is due to the reduction in FFV with addition of IL to PIL, which is only moderately improved by further supplementary $Zn[Tf_2N]_2$. Although some deviations are observed for **P9IL6Zn1** and **P9IL6Zn9**, all data points fit the linear Henry model well or the dual-mode sorption model at low pressures, where $b \cdot p << 1 + b \cdot p$, and thus the $CO_2$ concentration or the $CO_2$ sorption capacity increases with increasing $CO_2$ partial pressure. These results confirm the solubility governed $CO_2$ transport in PIL/IL/$Zn^{2+}$ composites.

ESI Figure S4 reveals that $CO_2$ sorption capacities of all investigated materials decrease with increasing temperature. Importantly, the Henry's constant $k_D$ decreases slightly with increasing temperature, following the Arrhenius equation up to approximately 50 °C (ESI Figure S5). This is in agreement with the constant material–gas system composition if assumed that partial molar enthalpy and entropy of the penetrant are not temperature dependent [77].

The gas transport through the dense membranes synthesized from **P9IL0Zn0**, **P9IL6Zn0**, **P9IL6Zn1** and **P9IL6Zn9** follows the solution–diffusion mechanism. ESI Table S1 reports the permeability ($P$), diffusion ($D$) and solubility ($S$) coefficients and the respective ideal selectivities with respect to nitrogen ($P_x/P_{N_2}$; $D_x/D_{N_2}$; $S_x/S_{N_2}$). Time lag experiments using single gases showed that when IL was added to PIL (**P9IL6Zn0**), the $CO_2$ permeability increased by 22 times, raising to 166 Barrer from 7.6 Barrer

for ($P_9IL_0n_0$). This behavior is due to the faster diffusion of $CO_2$ molecules through the less viscous segments of IL and is in accordance with previous reports [16]. Further $Zn[Tf_2N]_2$ salt incorporation in the PIL/IL composite counteracted the advantage of IL presence resulting in a lower permeability in the two PIL/IL/$Zn^{2+}$ composites of 80.9 Barrer and 20.2 Barrer for $P_9IL_6Zn_1$, and $P_9IL_6Zn_9$, respectively.

The ideal $CO_2/N_2$ permeability selectivity reveals an opposite trend compared to the one estimated from the gravimetric sorption experiments. The values obtained in the time-lag measurements increase in the following order: $P_9IL_6Zn_1 < P_9IL_0Zn_0 < P_9IL_6Zn_9 < P_9IL_6Zn_0$. In the case of $P_9IL_0Zn_0$ $D_{N_2} > D_{CO_2}$ which is unusual for glassy polymers and supports the discussion about the gravimetric sorption results in Section 3.2. These results suggest that although the total transport of $CO_2$ molecules was solubility controlled, the stronger $N_2$ solubility in $P_9IL_6Zn_1$ and slower $N_2$ diffusion in $P_9IL_6Zn_9$ affected the overall permeability performances of the composite materials.

The permeability properties of the pure PIL are closer to the properties of typical glassy polymers. This is visible in the He/$CO_2$ selectivity, since He permeates more than $CO_2$ in the $P_9IL_0Zn_0$-based membranes, showing a permeability selectivity of 1.5. In the membranes prepared from PIL/IL/$Zn^{2+}$ composite materials, the He/$CO_2$ selectivity is reversed, since He permeates less than $CO_2$. This indicates that the composite materials are similar to rubbery polymers, wherein the transport is "solubility controlled."

Figure 6 compares the ideal separation performances of investigated materials with the reported data for three gas pairs: $CO_2/N_2$, $CO_2/CH_4$ and He/$CO_2$ [78]. The composite materials containing [Pyrr$_{14}$][Tf$_2$N] and $Zn[Tf_2N]_2$ salt are positioned in the central part of the plot. $P_9IL_6Zn_1$ and $P_9IL_6Zn_9$ reveal moderate separation efficiency, while $P_9IL_6Zn_0$ shows the best performance enhanced by the presence of low viscosity IL segments.

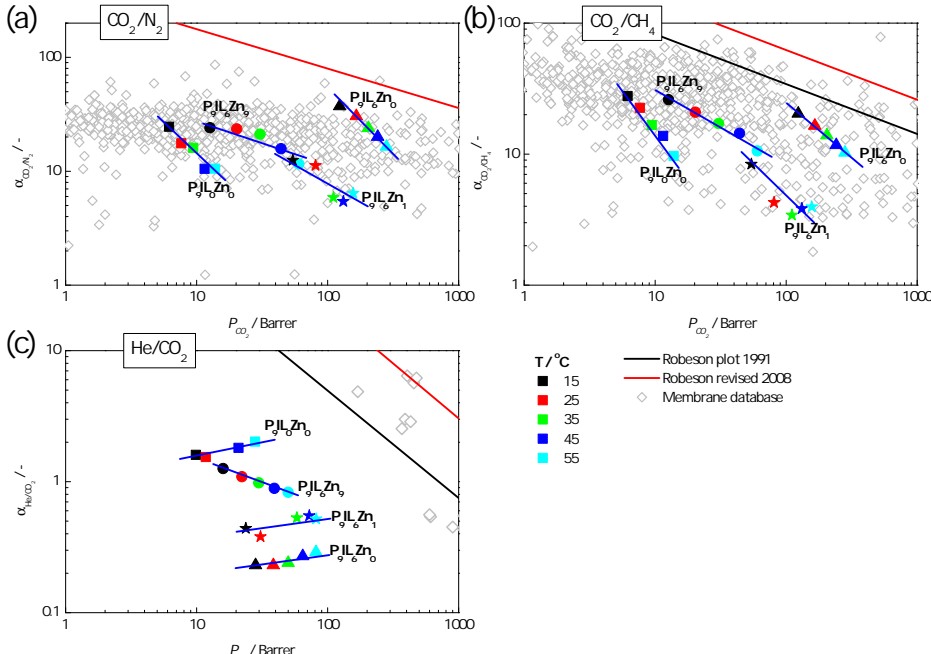

**Figure 6.** Positions of the composite PIL/IL/$Zn^{2+}$ materials on the $CO_2/N_2$ (**a**), $CO_2/CH_4$ (**b**) and He/$CO_2$ (**c**) Robeson's plots. The ideal separation performances of reported membrane materials are presented for comparison and are freely available from the Membrane Society of Australasia [78]. Blue solid lines are for guiding the eyes.

### 3.4. Thin-Film Composite Membranes Morphology

SEM images reveal the desired TFC membrane morphology (Figure 7). The morphology of the pure PIL **$P_9IL_0Zn_0$**-based membrane is comparable to the previously reported one [16]. The approximate thicknesses of the selective layer and the sealing PDMS layer were estimated from SEM images and are indicated on the corresponding SEM image. The average selective layer thicknesses were between 5–10 µm.

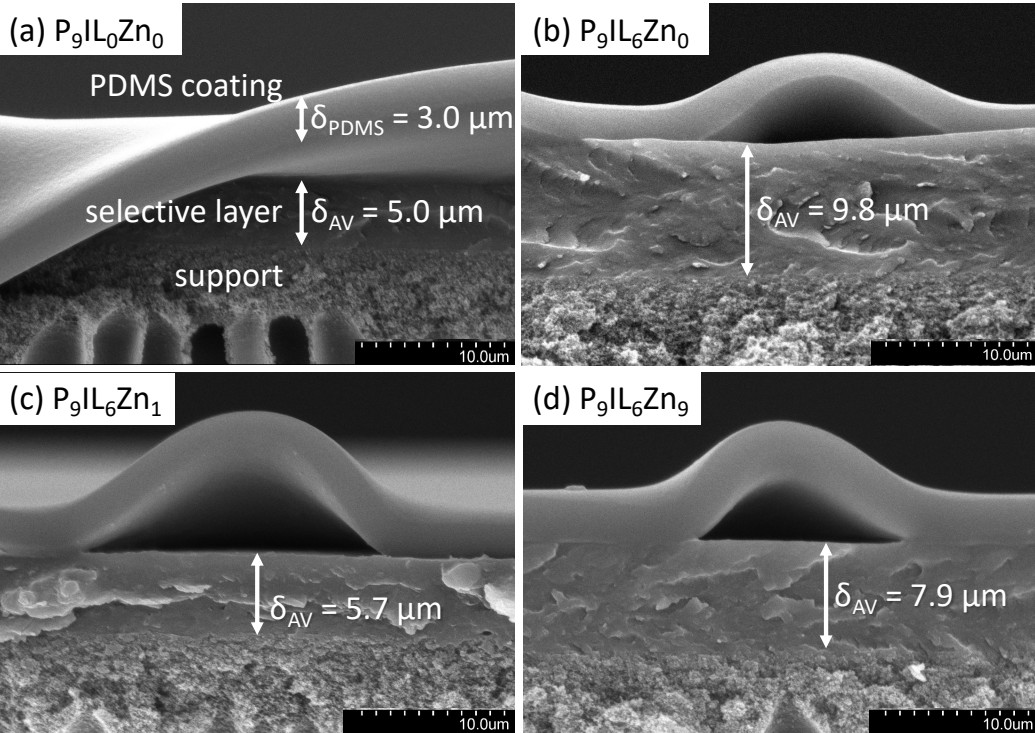

**Figure 7.** Cross-sectional SEM images depict the composite layered morphology of the membranes: (**a**) **$P_9IL_0Zn_0$**, (**b**) **$P_9IL_6Zn_0$**, (**c**) **$P_9IL_6Zn_1$**, (**d**) **$P_9IL_6Zn_9$**. Arrows indicate the corresponding layer thicknesses of PDMS sealing and selective layers.

For all TFC, the selective layer lies on top of the PI support (Figure 7). The interface between the selective layer and the support is characterized by gradual textural changes from the smooth dense coating layer to a rougher support. This behavior represents a good adhesion between the PIL/IL/$Zn^{2+}$ composites and the PI support, which also prevents the penetration of the selective layer in the porous matrix of the support. The PDMS sealing layer is resting on the selective material film and has a thickness of approximately 3.0 µm. The transition to the PDMS coating is sharper. The voids under the PDMS coating are probably the result of mechanical damage during fracturing of the sample. Ease of processing the composites into robust TFC membranes by simple solvent-casting method potentially allows their further consideration in flue gas separation applications. However, for the effective exploitation of these materials, improved technical solutions such as the use of a gutter layer and a further reduction of the selective layer thickness should be investigated.

### 3.5. TFC Membrane Performance in Conditions Imitating Flue Gas Separation

The high affinity of PIL/IL/$Zn^{2+}$ materials to $CO_2$ was found to be maintained for a wide range of $CO_2$ partial pressures and temperatures, further supporting the hypothesis that $CO_2$ molecules could be faster adsorbed in the membrane. Hence, the TFC membranes with s PIL/IL/$Zn^{2+}$-based selective layer may be deployed as a robust separation method for $CO_2$ removal from different gas streams, particularly flue gas streams with high relative humidity. Figure 8 represents the measured $CO_2$ permeance and $CO_2/N_2$ mixed-gas selectivity values in dry conditions (a) and in 90 % RH conditions at

variable feed pressures for pure P[DADMA][Tf$_2$N] (**P$_9$IL$_0$Zn$_0$**) and PIL/IL/Zn$^{2+}$ composite materials (**P$_9$IL$_6$Zn$_0$**, **P$_9$IL$_6$Zn$_1$** and **P$_9$IL$_6$Zn$_9$**) based membranes.

### 3.5.1. Influence of Relative Humidity

In dry conditions (Figure 8a), the CO$_2$ permeance moderately decreases with increasing feed pressure following the DMM sorption model. This behavior confirms the solubility-controlled CO$_2$ transport for all investigated materials, except for **P$_9$IL$_6$Zn$_1$**, the CO$_2$ permeance of which remains stable over the whole range of applied pressures. Importantly, CO$_2$ permeance in all blend membranes is at least 50 % higher when compared to pure PIL (**P$_9$IL$_0$Zn$_0$**). The CO$_2$/N$_2$ selectivity improves at higher feed pressures for all membranes except for **P$_9$IL$_6$Zn$_1$**. The exceptional behavior of **P$_9$IL$_6$Zn$_1$** correlates well with the solubility driven separation process, as the solubility of N$_2$ increases rapidly with increasing partial pressure of N$_2$ (1–6 bar range) according to Figure 3b. **P$_9$IL$_6$Zn$_0$** and **P$_9$IL$_6$Zn$_9$** achieve $\alpha_{CO_2/N_2}$ values comparable to the parent PIL material at sensibly higher overall permeance.

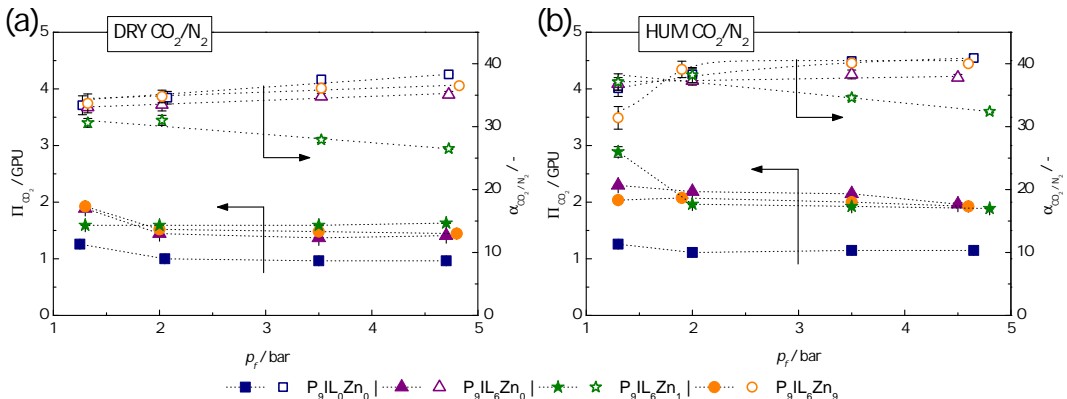

**Figure 8.** Influence of the feed pressure on the separation performance of PIL/IL/Zn$^{2+}$ composite-based TFC membrane at 26 °C in dry (**a**) and humidified (**b**) conditions of the sweep test set-up. The relative humidity of feed gas was ca. 90 % for humidified conditions.

In humidified conditions (Figure 8b), the CO$_2$ permeance shows a different trend in comparison to the dry conditions. For pure PIL, CO$_2$ permeance hardly changed, while the composite materials show a two-fold permeance increase from the initial value observed for the parent PIL. This suggests that humidified conditions not only provide additional transport mechanism for CO$_2$ molecules, referred to as internal sweep [33], but also allow water molecules to adsorb in the material matrix, plasticize it and make it more rubber-like [17].

Remarkably, 90 % relative humidity enhances $\alpha_{CO_2/N_2}$ for all investigated materials by at least 10 %. Here again, **P$_9$IL$_6$Zn$_9$** exhibits a comparable CO$_2$/N$_2$ selectivity to pure PIL, while permeating two times more CO$_2$ over the broad range of feed pressures. This phenomenon may signify that in a hydrated state, the IL and Zn[Tf$_2$N]$_2$ constituents play an even more significant role in CO$_2$ sorption. This observation supports the hypothesis that incorporation of additives containing [Tf$_2$N]$^-$ anions, strongly interacting with CO$_2$ molecules, and Zn$^{2+}$ cations, provides additional CO$_2$ separation benefits by encouraging the facilitated transport mechanism in the PIL/IL composites.

### 3.5.2. Influence of Temperature

Figure 9 represents the measured CO$_2$ permeance and CO$_2$/N$_2$ mixed-gas selectivity values in dry conditions (a) and in 90 % RH conditions (b) at 1.2 bar and varied process temperatures for pure P[DADMA][Tf$_2$N] (**P$_9$IL$_0$Zn$_0$**) and PIL/IL/Zn$^{2+}$ composite materials (**P$_9$IL$_6$Zn$_0$**, **P$_9$IL$_6$Zn$_1$**, and **P$_9$IL$_6$Zn$_9$**) based membranes. Both in dry and in humidified process conditions, CO$_2$ permeance increased, and CO$_2$/N$_2$ mixed-gas selectivity decreased at higher temperatures. In dry conditions,

the performance trends for all materials were similar, while the pure PIL-based membranes (**P₉IL₀Zn₀**) broke at increased process temperatures (> 30 °C).

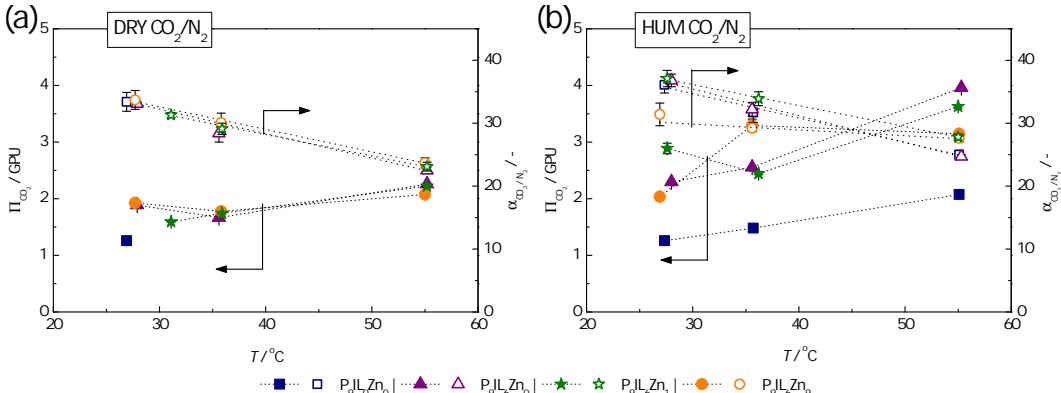

**Figure 9.** Influence of the temperature on the separation performance of PIL/IL/Zn²⁺ composite-based TFC membrane at 26 °C, 35 °C and 55 °C and 1.2 bar in dry (**a**) and humidified (**b**) conditions of the sweep test set-up. The relative humidity of feed gas was ca. 90 % for humidified conditions.

In humidified conditions, the pure PIL-based membrane continued to perform in a way resembling the performances of composite materials-based membranes in the dry process conditions. This suggests that a minimum humidity in the membrane matrix is required to preserve the $CO_2$ transport. Additionally, the plasticization effect of water vapor might improve the matrix stability of the PIL by decreasing its rigidity. The composite materials showed a two-fold permeance increase from the initial value observed for the parent PIL. This confirms the results obtained in the tests with varied feed pressures.

Humidity presence in the feed stream also positively affected $\alpha_{CO_2/N_2}$ for all investigated materials with an increase of least 10 %. $CO_2$ permeance and $\alpha_{CO_2/N_2}$ were most stable for **P₉IL₆Zn₉** over the broad range of tested temperatures. This observation supports the idea that while the sorption of $CO_2$ in other materials considerably decreases with increasing temperature, the **P₉IL₆Zn₉** strong affinity towards $CO_2$ molecules in Langmuir sites compensates these losses. Thus, these results provide a sensitive measure of **P₉IL₆Zn₉**-based membrane robustness in terms of process conditions in a wide range of feed pressures, temperatures and humidities imitating the flue gas $CO_2$ capture.

## 4. Conclusions

Effective $CO_2$ selective materials prepared from commercially available P[DADMA][Cl] were successfully converted in poly(ionic liquid)-based composite materials with improved $CO_2$ sorption properties. Incorporation of [Pyrr₁₄][Tf₂N] and Zn[Tf₂N]₂ considerably altered the ideal separation performances of the composites. The time-lag experiments show that the addition of the IL improves the $CO_2$ diffusivity in the PIL/IL composite. Furthermore, the supplementary [Tf₂N]⁻ anions and Zn²⁺ cations, supposedly interacting with $CO_2$ molecules, enhance the sorption capacity of the P₉IL₆Zn₉ composite. Although certain deviations between the results from direct and indirect solubility, diffusivity and permeability data are identified in the gravimetric and the time-lag experimental approaches, the results obtained capture the dominating contribution of $CO_2$ sorption, and this means that the samples exhibit a predominantly solubility-controlled transport mechanism. PIL/IL/Zn²⁺-based TFC membranes remain functional in a broad range of process conditions investigated, imitating realistic flue gas $CO_2$ capture. Importantly, the improved mixed gas separation performance of composite materials in the humidified conditions demonstrates a combination of transport mechanisms: solution-diffusion and facilitated transport.

**Supplementary Materials:** The following are available online at http://www.mdpi.com/2076-3417/10/11/3859/s1.

**Author Contributions:** Conceptualization, D.N.; methodology, D.N.; validation, D.N.; formal analysis, M.T. and A.F.; investigation, D.N., S.L., M.S. and P.I.D., J.J., J.C.J. and M.T.; resources, I.F.J.V., J.C.J. and P.I.D.; data curation, D.N.; writing—original draft preparation, D.N.; writing—review and editing, D.N., M.T., A.F. annd J.C.J.; visualization, D.N.; project administration, D.N.; funding acquisition, I.F.J.V. All authors have read and agreed to the published version of the manuscript.

**Funding:** The financial support of this project by the European Union Seventh Framework Programme FP7/2007-2013 under grant agreement number 608535 is gratefully acknowledged. The authors are also grateful for the financial support from the OT (11/061) funding from KU Leuven, the Belgian Federal Government through I.A.P.- P.A.I. grant (IAP 7/05 FS2), the Polish Ministry of Science and Higher Education through International Co-financed Project grant (2978/7.PR/2014/2) and the Erasmus Mundus fellowship funded by EACEA (EUDIME doctoral programme 4th edition).

**Acknowledgments:** Authors acknowledge Edel Sheridan (SINTEF), Jannicke Hatlø Kvello (TGA, clean room, SINTEF) and Julian Richard Tolchard (SEM/SINTEF) for conducting the mentioned measurements and helping with their interpretation.

**Conflicts of Interest:** The authors declare no conflict of interest.

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
