# Peer review of "Water Vapour Promotes CO2 Transport in Poly(ionic liquid)/Ionic Liquid-Based Thin-Film Composite Membranes Containing Zinc Salt for Flue Gas Treatment"

_applsci, doi:10.3390/app10113859_

Round 1
Reviewer 1 Report
The aim of this paper is to synthesize thin film composite membranes based on poly(ionic liquid)-ionic liquid selective layers that contain zinc salts and characterize the CO2 permeation properties under humid conditions. To me, the topic is very relevant in the field as novel IL-based TFC membranes are synthesized, which is the required next step for further development of IL-based membrane technology. The manuscript is very well-written and the authors provide a nice description of the membrane features. Therefore, I highly recommend its publication in the journal.
Here is a list of minor comments for their consideration:
Possible typo in page 2 line 67: “or ¿to? too strong coordination”
In Table 1, specify units of %, mol or wt?
Permeance results of all TFC-membranes are around 2 GPU for CO2 (dry conditions). With selective layers around 5-10 microns, these numbers would approximately yield 10-20 barrer, which is much lower than the permeability obtained through the thick films. If this is correct, what do the authors think about this significant reduction in permeability when moving from thick films to TFC? Would it hamper the development of IL-based TFC membranes?
Reviewer 2 Report
This manuscript contains experimental studies on CO2 transport in Poly(iconic liquid) and Ionic liquid-based thin-film composite membrane containing Zinc. The authors employ solubility measurements, and other technniques like SEM, TGA, and DSC. The interest in CO2 separation and capture has increased in recent years due to concerns about the impact of such emissions on the environment, so this paper is quite topical. In addition, its focus on the impact of water vapor on transport properties is of interest to those working with ionic liquids. The paper contains interesting results. Below are some comments/suggestions on how to improve on its quality:
- Minor Comment: Lines 1-4 in the Abstract are more appropriate to be used in the Introduction part of the paper, augmented with the appropriate references.
- Minor Comment: The FFV acronym is used for the first time on line 238 without having been previously defined on what it stands for.
- Major Comment: The Figures included in the electronic supporting information (ESI) section are referenced several times and elaborated in great detail in the paper. The authors need to reconsider whether some of these figures should be, in fact, included in the manuscript itself. If there are integral to the discussion in the paper, and are referenced multiple times, there is really no good reason for delegating them to the ESI section of the paper.
- Major Comment: In Figure 1 in the ESI section, while comparing parts (a) and (c) it is noted that the peaks occurring in part (a) were due to traces of humidity. Can you provide any literature reference that supports such a conclusion? What were the conditions of the ESI Figure 1c that ensured that the humidity was eliminated and how many times was the test repeated.
- Major Comment: The experiments for the sorption behavior of CO2 and N2 using the gravimetric method lasted 360 min. The samples P9IL6Zn0, P9IL6Zn1, P9IL6Zn9 contain IL that is in a gel phase, therefore, the time that they may need to reach equilibrium with the gas is likely to be higher than 360 min. The authors, themselves, also commented in the paper that sorption equilibrium may require longer time. The question then is, are the results reported in this paper obtained under such condition reliable? Experimental validation is needed, that 360 min are sufficient for sorption equilibrium to be established.
Reviewer 3 Report
The MS entitled „Water vapour promotes CO2 transport in Poly(ionic liquid) / Ionic Liquid-based Thin-film Composite Membranes containing Zinc Salt for Flue Gas Treatment“ written by Daria Nikolaeva, Sandrine Loïs, Paul Inge Dahl, Marius Sandru, Jolanta Jaschik, Marek Tanczyk, Alessio Fuoco, Johannes Carolus Jansen, and Ivo F.J. Vankelecom focuses on membranes made of polydiallyldimethyl ammonium NTf2, N-butyl-N-methyl pyrrolidinium NTf2, and Zn NTf2 as well as their investigation for gas separation (CO2 and N2). The MS is interesting. However, some improvements are necessary before publication.
DSC curves of various samples are compared with each other, although the DSC curve of the pure ionic liquid is missing in this MS whereas the DSC curve of the pure PIL (P9IL 0 Zn0) is given in the Supporting Information. Comparison of the composites with the pure substance is highly recommended within this manuscript.
Authors pointed out that humidity influenced their results several times. However, they did not give any number for the humidity.
Furthermore, information on page 8 “These anomalies give an impression that some measurement points at lower pressure range might have had insufficient
time to equilibrate, affecting their accuracy.” Repeating of these experiments may be helpful to make this effect more clear.
Moreover, the authors pointed out that “The IL domains have lower viscosity within the material matrix …” (page 8). If IL domains exist in the membranes, the DSC curves of the membranes should show the transition temperature of the IL. Therefore, I recommend to compare the DSC curve of the pure IL with those of the membranes.
Conclusions are just a summary. Therefore, the conclusions need improvement as well.
I cannot recommend the MS in the present form.
Reviewer 4 Report
In this study the authors prepare composite membranes by blending a bistriflimide-containing ionic polymer with a bistriflimide ionic liquid and Zn(NTf2)2. The polymer/IL composites have higher diffusion coeffecients and permeability for CO2 and for nonpolar gasses consistent with their greater fluidity. CO2 permeance increases selectively for Zn2+-impregnated membranes under humid and high temperature conditions, which is attributed to multiple phenomena such as the plasticization of the gel by H2O vapor and affinity of the Zn2+ sites for CO2 compensating for decreased gas affinity at high temperatures.
The manuscript is well written, differs significantly from the authors’ other work in the area, and provides a useful literature background on the strategy of modifying polymerized ILs. I recommend only a few minor revisions to better describe the experimental details.
More detail needs to be added regarding the blending of the polymer and the additives. Physical blending can encompass a wide range of techniques ranging from hand-blending using a mortar and pestle to any of a number of automated milling devices. The description of the materials after blending should be added here as well; apparently they are gels according to the footnote of Table 2.
Viscosity measurements would be a great addition, both for the casting solutions used to make the membranes and for the composite itself if it is in a suitable form. Since the composites are made by blending two solids with a viscous, largely nonpolar ionic liquid, it is highly likely that changes in the blending procedure will alter the composition of the final material. Characterizing the casting solutions by both viscosity and weight percent could greatly aid reproducibility since viscosity is an intrinsic property that is probably both the easiest to measure and the most directly connected with the material properties of the resultant membrane.
TGA of Zn(NTF2)2 would be helpful in confirming whether the loss of thermal stability is due to the effect of Zn2+ on the material or simply because Zn(NTf2)2is not stable at those temperatures.
